# Cervical Secretions from Women After Depot Medroxyprogesterone Acetate (Depo-Provera) Administration Promote HIV Infectivity Ex Vivo

**DOI:** 10.3390/v17091283

**Published:** 2025-09-22

**Authors:** Carley Tasker, Natalie E. Roche, Yungtai Lo, Theresa L. Chang

**Affiliations:** 1Public Health Research Institute, Rutgers, New Jersey Medical School, 225 Warren Street, Newark, NJ 07103, USA; cctasker@gmail.com; 2Department of Microbiology, Biochemistry and Molecular Genetics, Rutgers, New Jersey Medical School, Newark, NJ 07103, USA; 3Department of Obstetrics, Gynecology & Women’s Health, Rutgers, New Jersey Medical School, Newark, NJ 07103, USA; rochene@njms.rutgers.edu; 4Department of Obstetrics, Gynecology and Reproductive Health, University Hospital, Newark, NJ 07103, USA; 5Department of Epidemiology & Population Health, Albert Einstein College of Medicine, Bronx, NY 10461, USA; yungtai.lo@einsteinmed.edu

**Keywords:** Depo-Provera, vaginal and cervical mucosa, HIV infectivity

## Abstract

Depot medroxyprogesterone acetate (Depo-Provera) has been associated with an increased risk of HIV acquisition. We have previously shown that Depo-Provera administration increases immune markers for HIV preference on peripheral and cervical CD4^+^ T cells but decreases the levels of most immune mediators at vaginal and cervical mucosa. In this study, we determined the effect of cervicovaginal secretions from women before (visit 1), one month (visit 2) and three months (visit 3) after Depo-Provera treatment on HIV infectivity ex vivo. The effect of supernatants from vaginal, endocervical, and rectal swabs and from cervical cytobrush on HIV infectivity were assessed by a single-cycle infection assay using CCR5-using HIV-luciferase reporter viruses. We found that endocervical secretions from women after Depo-Provera treatment promoted HIV infectivity. When analyzing the association between endocervical mediator changes in response to Depo-Provera, available in our previous study, and the changes in HIV infectivity pre- and post-treatment, we found that changes in IL-17 and VEGF were positively associated with changes in HIV infectivity at visit 2 compared with visit 1, whereas changes in RANTES and IL-4 were negatively associated with HIV infectivity. The negative association between RANTES and HIV infectivity was also observed at visit 3 compared with visit 1. Additionally, changes in IL-1α at visit 3 were positively associated with changes in HIV infectivity compared with visit 1. These findings suggest that Depo-Provera may increase the HIV risk by shifting the mucosal milieu that promotes HIV infectivity.

## 1. Introduction

Depot medroxyprogesterone acetate (Depo-Provera), an injectable hormonal contraceptive administrated every 3 months, has been associated with an increased risk of HIV acquisition (reviewed in [1,2]), although some studies, including the Evidence for Contraceptive Options and HIV Outcomes (ECHO) trial of more than 7800 women, reported no significant difference in HIV risk among women using Depo-Provera [3,4]. The ECHO trial compared HIV risks among the users of Depo-Provera, the levonorgestrel implant, and a copper intrauterine device, but women without using hormone contraceptives were not included in the study [3]. While the results of the ECHO trial led the World Health Organization to relax its guidelines for women at risk, the association between Depo-Provera use and HIV risk remains debated [5,6,7,8,9,10].

We previously conducted a longitudinal study by collecting vaginal, endocervical, and rectal swabs as well as cervical cytobrush and blood from 39 women at baseline (visit 1, V1), 1 month (visit 2, V2), and 3 months (visit 3, V3) after Depo-Provera, and analyzed immune cells by flow cytometry and immune mediators by multiplex assays and ELISA [11,12]. Our results showed that Depo-Provera administration increased the frequency and expression levels of immune markers for HIV preference including integrin α4β7, CCR5, and CD38 on peripheral CD4^+^ T cells, and promoted susceptibility of PBMCs to HIV infection ex vivo [11]. At the cervix, we and others found that the use of Depo-Provera increases the numbers of cervical CCR5^+^CD4^+^ T cells [12,13,14]. Analysis of the levels of immune mediators in supernatants of the cervical and vaginal specimens showed that Depo-Provera injection resulted in general immune suppression [12]. For example, in the cervical supernatants from cytobrushes, the levels of 22 (out of 25) cytokines/chemokines from V2 and the levels of 19 mediators from V3 were significantly decreased compared with V1 (pre-treatment) [12]. Immune profiles differed between vaginal and cervical mucosa in response to Depo-Provera, indicating immune compartmentalization at the lower female reproductive tract (FRT) [12]. Interestingly, EGF was significantly induced in the vaginal secretion at V3 and IL-17 was significantly induced in the endocervical secretion at V2 compared with V1 [12]. Additionally, there is a non-significant increase in RANTES at V2 in both cervical swab and cytobrush samples.

The effects of Depo-Provera on the levels of immune mediators in the FRT have not been consistent in cross-sectional studies [2]; however, the overall immune suppressive profile in response to Depo-Provera in our study is consistent with a longitudinal study of 15 women over a 12-month period which analyzed IL-6, IL-8, and IL-1RA in vaginal secretions [15]. Other cross-sectional studies also found cervical immune suppression in Depo-Provera users [16,17].

While the changes in immune milieu in response to Depo-Provera administration have been studied extensively, the effect of cervicovaginal secretions from women pre- and post-Depo-Provera administration on HIV infectivity ex vivo has not been determined. In this study, we used cervicovaginal and rectal secretions from women pre and post Depo-Provera administration, which were collected in our previous study [12], to determine the effect of clinical samples on the infectivity of CCR5-using virus HIV-1_JR-FL_-luciferase reporter virus. Our result showed that the endocervical secretions from women after Depo-Provera administration promoted HIV infectivity compared to samples at the baseline.

## 2. Materials and Methods

### 2.1. Study Subjects and Sample Collection

The study (Pro20140000011) was approved by Rutgers, New Jersey Medical School (NJMS) Institutional Review Board (date approved: 30 October 2024). The Helsinki Declaration has been followed and written informed consent were obtained from the subjects for this study. Women who met the eligibility criteria were recruited at Rutgers, NJMS clinics located in Newark, New Jersey, as described in our previous study [11,12]. Briefly, healthy, non-pregnant women aged 18–35 without known or suspected HIV or bacterial sexually transmitted infections, cervical carcinoma, or cancer were eligible. Subjects had no sexual intercourse for 3 days before enrollment visit. Samples were collected prior to Depo-Provera injection (visit 1, V1), one month (visit 2, V2), and three months following the initial Depo-Provera injection prior to the second injection (visit 3, V3). Cervical cell samples were collected by sweeping the endocervix with a Medscand Cytobrush Plus GT (Cooper Surgical), which was then immersed in 3 mL of RPMI-1640 without serum. Supernatants were aliquoted and stored at −80 °C. Vaginal, endocervical, and rectal swabs were collected using sterile HydraFlock swabs (Puritan), and then placed in 0.7 mL of PBS. Supernatants were collected, aliquoted, and stored at −80 °C. Prostate-specific antigen (PSA), a marker for the presence of semen, was measured by ELISA (R&D, cat#DKK300). PSA positive samples, indictive of recent sexual intercourse, were not included in the study. Among 67 recruited subjects, data from 39 participants who met the eligibility criteria are presented in this study. Samples were collected between May 2015 and March 2017. All subjects were Black or Hispanic reflecting the demographic characteristics of the clinical site in Newark, NJ, USA. The median age was 26 [interquartile range (IQR), 21.5–30.5]. Subjects had not used any form of hormonal contraception for up to 10 months at the time of recruitment. The immune mediators were analyzed by multiplex assays and ELISA as described in our previous study [12].

### 2.2. HIV Infectivity Assay

The effect of mucosal secretions on HIV infectivity was assessed in a single-cycle infection assay as described previously [18]. Vaginal, endocervical, and rectal secretions from swabs as well as cervical cytobrush supernatants were incubated with serum-free pseudotyped HIV luciferase reporter virus with CCR5-using HIV-1_JR-FL_ envelope for 1h at 37 °C. The mixture of virus-mucosal secretions was added to HeLa-CD4-CCR5 cells (JC48 cells), provided by David Kabat at Oregon Health and Science University, Portland, Oregon, for 2 h at 37 °C. After washing off unbound viruses, infected cells were cultured in DMEM media with 10% FBS, Amphotericin B, and gentamicin (Sigma-Aldrich, St. Louis, MO, USA) for 2 days before lysis with passive lysis buffer (Promega, Madison, WI, USA). PBS-treated viruses were included as a control in each plate. Luciferase activity (in relative light units, RLUs) was read using a 2300 EnSpire Multilabel Plate Reader (PerkinElmer, Waltham, MA, USA). Luciferase values of clinical sample-treated viruses were normalized to virus incubated with PBS alone. To summarize the results from multiple donors with different visits that were analyzed in triplicates in multiple plates, the average percentage of control was calculated using the following formula: (mean RLUs of treated samples (triplicate)/mean RLUs of PBS (triplicate)) × 100%. The value of viral infectivity in PBS is 100%.

### 2.3. Statistical Analysis

Wilcoxon matched-pairs signed-rank test was used to compare differences in HIV infectivity between study visits. Spearman’s rank-order correlation for the endocervical samples was used for correlation analysis of changes in HIV infectivity (e.g., the value at V2 or V3 was subtracted by the value at V1), and changes in cytokine levels in samples pre vs. post Depo-Provera treatment (V2-1V1, V3-V1). Statistical analysis was performed using GraphPad Prism version 8.4.2 (GraphPad Software, Inc.
San Diego, CA, USA); *p* < 0.05 was considered statistically significant.

## 3. Results and Discussions

Vaginal, endocervical, and rectal secretions from swabs and cytobrush supernatants from women before (visit 1, V1), one month following (visit 2, V2), and three months following (visit 3, V3) Depo-Provera injection were collected [12]. The three-month schedule was chosen because serum Depo-Provera concentrations are known to decrease over time and return to near baseline three months after Depo-Provera injection [19,20]. To address whether pre- and/or post-Depo-Provera treatment vaginal, endocervical, or rectal secretions or cytobrush supernatants from subjects pre-and post-Depo-Provera affected HIV infectivity, we determined the effect of mucosal secretions on HIV infectivity in a single-cycle infection assay. Note that the assay was designed to address the effect of secretions on viral entry as mucosal secretions were washed off after viral attachment. The infectivity data with clinical samples were normalized based on PBS-treated samples in the assay.

Among samples from different compartments, endocervical secretions from women after Depo-Provera treatment promoted HIV infectivity compared to pre-treatment samples (Figure 1). There was no significant difference in HIV infectivity when viruses were treated with vaginal, rectal and cytobrush samples from women with pre- and post-Depo-Provera treatment. Specifically, there was an increase in HIV infectivity when viruses were pre-treated with endocervical secretions from V3 (median 199.1%, IQR 112–258.9%) compared to samples from V1 (median 95.4%, IQR 65.9–151.6%) or compared to samples from V2 (median 125.6%, IQR 92.7–154.8%). There was no change between vaginal secretions from V1 (median 161%, IQR 96.6–261.8%) and V2 or between V1 and V3, although vaginal secretions from V3 (median 221.4%, IQR 121.9–345.2%) exhibited a significant increase in HIV infectivity compared to samples from V2 (median 135%, IQR 92.5–211%).

The effect of rectal secretions on HIV infectivity was not significantly affected by Depo-Provera treatment (V1: median 191.3%, IQR 112.7–327.9%; V2: median 203.5%, IQR 119.1–295.5%; V3: median 236.2%, IQR 97.4–362%). Supernatants from cytobrush at V2 or V3 increased HIV infectivity compared to samples at V1 (V1: median 45.1%, IQR 23.9–94.8%; V2: median 77.3%, IQR 33.2–121.2%; V3: median 67%, IQR 27.9–119.6%), although the increase did not reach significance. The less pronounced Depo-Provera-induced HIV enhancement in the cytobrush supernatants may be due to cytobrush samples that were more diluted (3 mL) compared to samples from swabs (0.7 mL).

Because endocervical secretions from women with Depo-Provera promoted HIV infectivity ex vivo compared to samples at V1 (pre-treatment), we determined whether the changes in HIV promoting activity after Depo-Provera treatment could be associated with changes in endocervical immune mediators that were determined in our previous study [12]. When compared changes between V1 and V2, we found that changes in IL-17 (*p* = 0.016, *r* = 0.3833) and VEGF (*p* = 0.0035, *r* = 0.5164) had a moderate association with changes in HIV infectivity, whereas RANTES (*p* = 0.0003, *r* = −0.6374) had a significant negative association with HIV infectivity (Table 1). IL-4 (*p* = 0.0493, *r* = −0.1287) also had a weak negative association with changes in HIV infectivity. Except for RANTES (*p* = 0.0131, *r* = −0.4797), the association of changes in these cytokines with changes in HIV infectivity were not found in V3 compared with V1. In addition to RANTES, changes in IL-1α (*p* = 0.05, *r* = 0.3803) had a moderate association with changes in HIV-1 infectivity at V3 compared with V1.

IL-17 is critical for maintaining mucosal immunity against viral infection but can also promote HIV infection and pathogenesis [21,22,23]. HIV exposed, seronegative (HESN) sex workers have reduced levels of IL-17 in the cervix and blood compared to non-sex workers, indicating that lower levels of IL-17 are associated with HIV resistance at the FRT of HESN subjects [24]. High levels of IL-17 are associated with high plasma viral loads in HIV-1 subtype C patients without anti-retroviral therapies (ARTs) [25]. Additionally, a study with 18 HIV-1 chronically infected men off ARTs shows that the seminal plasma HIV-1 RNA level is positively associated with IFN-γ and IL-17 and negatively associated with Th2 cytokine IL-5 in seminal plasma [26]. Similarly, we found that IL-17 was positively associated with increased HIV infectivity, whereas the Th2 cytokine IL-4 was negatively associated with HIV infectivity. IL-17 is known to induce VEGF [27,28,29], which may explain the positive association of VEGF with increased HIV infectivity. Changes in IL-1α, a pro-inflammatory cytokine, were associated with increased HIV infectivity at V3 compared to V1, suggesting the effect of Depo-Provera persists. IL-1α has been associated with increased HIV risk in women [30].

RANTES can inhibit HIV entry via CCR5 [31] but promotes HIV entry via interacting with glycosaminoglycans when the cytokine concentration is high [32]. Here we found a negative association between RANTES and HIV infectivity, supporting its inhibitory role. RANTES was the only cytokine associated with ex vivo HIV infectivity at both V2 and V3 compared to V1. Interestingly, our previous report showed that cervical CCR5^+^CD4^+^ T cells, the preferred HIV target cells, were more abundant at V3 than at V1 [12], coinciding with reduced RANTES levels. Together, these findings suggest that decreased levels of the HIV entry inhibitor RANTES, along with an increased frequency of CCR5-expressing HIV-susceptible CD4^+^ T cells, may contribute to higher HIV risk in women receiving Depo-Provera.

We observed a significant difference in HIV infectivity between vaginal samples from V2 (one month) and V3 (three months), but not between V2 or V3 compared to V1. The absence of difference compared to V1 may be due to the lack of defining specific stages of menstrual cycles, which strongly influences vaginal tissues at the baseline. Depo-Provera treatment appeared to reduce the variability of HIV infectivity among vaginal samples at V2 compared to samples at V1, likely reflecting its early effects (one month) of Depo-Provera on the vaginal mucosa. Similarly, increased variation at V3 compared to V2 may be due to the reduced levels of medroxyprogesterone acetate (MPA) at V3. The degree of HIV infectivity was increased at V3, coinciding with the time subjects were due for their second Depo-Provera injection. In our previous report, epidermal growth factor (EGF) was the only mediator showing significant changes in the vaginal samples, increasing from V2 (median 26, IQR 16.17–64.15 pg/mL) and V3 (median 67.69, IQR 28.55–153.28 pg/mL) [12]. EGF has been reported to enhance HIV infectivity by promoting internalization of virions [33]. Although the low concentration of EGF in the vaginal samples are unlikely to fully explain the increased HIV infectivity at V3, its role of Depo-Provera-associated changes in the vaginal mucosa warrants further investigation.

Similarly, HIV infectivity differed between endocervical samples from V2 and V3. In our previous report, the cytokines that changed significantly between these visits were IL-7 (V2: median, 18.875, IQR, 12.03-20.65 pg/mL; V3: median, 13.07, IQR, 9.57–16.87 pg/mL) and RANTES (V2: median, 17.062, IQR, 8.41–35.16 pg/mL; V3: median 3.487, IQR 1.19–6.3 pg/mL) [12]. IL-7 plays a critical role in modulating T cell function but promotes HIV persistence [34,35,36]. For HIV transmission, IL-7 enhances HIV infection through multiple mechanisms including promoting viral replication in resting CD4^+^ T cells [37], increasing CD4^+^ T cell susceptibility by down-regulating SAMHD1 [38], and facilitating infection in cervical-vaginal tissues explants by regulating T cells [39]. Because the absence of CD4^+^ T cells in our HIV infectivity assays and the lack of increased IL-7 levels at V3 compared to V2, the mechanisms found in T cells cannot explain the potential role in increased HIV infectivity at V3. In contrast, RANTES is a potent HIV entry inhibitor and has been associated with increased HIV susceptibility and viral spread [40,41]. The decrease in RANTES may in part contribute to increased HIV infectivity at V3 compared to V2.

Taken together, it remains to be determined whether these cytokines have direct effects on viral entry or serve as markers for increased HIV risk. These markers together with HIV infectivity may be used as markers to predict HIV risk. Further biochemical studies are required to identify HIV promoting factors in secretions from Depo-Provera-treated women or HIV inhibitory factors from women before Depo-Provera injection.

In addition to its immunomodulatory function, Depo-Provera influences the epithelium at the FRT [42,43], mucin production [44], and vaginal microbiome [45,46], the latter two of which may directly affect ex vivo HIV infectivity assays. While Depo-Provera results in vaginal thinning and increased vaginal transmission of simian immunodeficiency virus [47], both cross-sectional and a longitudinal studies up to 3 months have not found apparent thinning of the vaginal epithelium in women [42,43], despite of its known thinning effect on the endometrium. IL-17 plays a critical role in maintaining epithelial barrier and supporting a balanced vaginal microbiome but can also promote inflammation and tissue damage, as observed in rheumatoid arthritis [21]. Bacterial vaginosis (BV)-associated metabolites inhibit RANTES induction in cervicovaginal epithelial cells in response to TLR2 [48]. We have previously shown that Depo-Provera treatment promotes a shift toward a BV-associated microbiome in Black women [45]. It remains to be determined whether the BV-associated metabolites following Depo-Provera treatment contribute to reduced RANTES and increased HIV infectivity at V3.

Depo-Provera increases cervical mucin thickness within 1–3 days of treatment, thereby reducing sperm penetration [44]. Mucins derived from human cervix or cervicovaginal mucosa have been shown to reduce HIV mobility in vitro [49], suggesting a potential protective role in preventing HIV penetration of the epithelial barrier and infection of target cells. However, only highly purified mucins isolated from cervical plugs but not crude cervical samples have demonstrated HIV inhibitory effects in vitro [50]. Given the critical role of mucins in maintaining the FRT function, assessing mucin levels and composition at baseline and one and 3 months after Depo-Provera treatment may provide additional insights into how Depo-Provera alters the cervicovaginal milieu that promotes host susceptibility to HIV.

Inflammatory milieu has been thought to contribute to increased HIV transmission at the FRT. Indeed, our previous study showed that chlamydia-infected women have higher levels of pro-inflammatory cytokines at the cervicovaginal mucosa than women after receiving antibiotic treatment, and endocervical secretions from chlamydia-infected women enhanced HIV infectivity ex vivo compared to samples from women post-treatment [18]. Although the overall immune suppressive profile at FRT from women with Depo-Provera administration in our previous study [12] might be expected to reduce HIV infectivity, we found the opposite, as endocervical secretions from women after Depo-Provera treatment promoted HIV infectivity ex vivo. These findings suggest that an overall pro-inflammatory cytokine profile is not universal markers for predicting susceptibility to HIV infection and that specific immune markers for predicating HIV risks may depend on specific clinical settings. Nevertheless, the ex vivo HIV infectivity assay may be a useful pre-clinical method to predict HIV risks.

The limitation of this study includes the absence of white subjects, as Black and Hispanic women were more receptive to using Depo-Provera at our clinics. Additionally, we did not define specific stages of menstrual cycles, which significantly influence vaginal tissues at V1. This may explain the lack of differences in HIV infectivity between Depo-Provera synchronized V2 or V3 compared with V1, which included participants at varying stages of menstrual cycles. Measuring the MPA levels in subjects at V2 and V3 could strengthen the analysis. It is possible that short-term Depo-Provera use promotes acute inflammation and the recruitment of target cells, whereas long-term use may sustain a chronic inflammatory environment, cause cumulative structural damage, and weaken the mucosal barrier. Therefore, analyzing samples from women with prolonged Depo-Provera use, increasing the sample size, and accounting for ethnicity and genetic factors may provide better insights into the effects of Depo-Provera on the FRT and their implications for HIV transmission.

## 4. Conclusions

In our 3-month longitudinal study, endocervical secretions from women after Depo-Provera treatment increased HIV infectivity ex vivo. Future investigations into the mechanisms by which Depo-Provera modulates and promotes HIV infectivity ex vivo will provide insights relevant to developing strategies to prevent HIV spread.

## Figures and Tables

**Figure 1 viruses-17-01283-f001:**
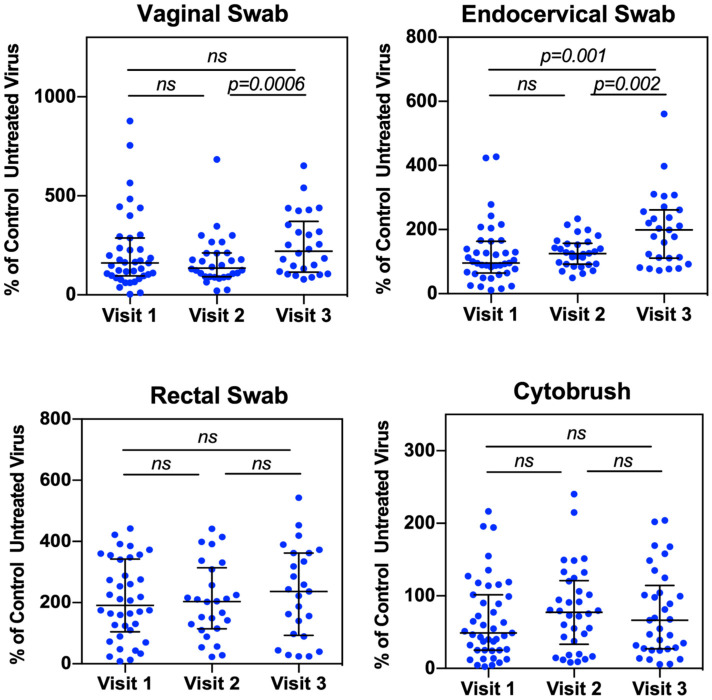
Vaginal and endocervical secretions from Depo-Provera treated patients enhance HIV infectivity. Vaginal, endocervical, and rectal secretions from swabs, and supernatants from the cytobrush were incubated with serum-free HIV-luc (JR-FL) for 1 h at 37 °C. Following incubation, samples were used directly to infect HeLa-CD4-CCR5 cells for 2 h at 37 °C. Following infection, cells were washed to remove unattached virus, and the level of HIV infection was determined 2 days later by luciferase assay. Luciferase values were normalized to virus incubated with PBS alone (untreated virus). Each dot represents an individual donor (median, IQR). A Wilcoxon matched-pairs signed-rank test was used for analysis. *p* ≤ 0.05 was considered significant. *ns*, not significant.

**Table 1 viruses-17-01283-t001:** Association of changes in immune mediators with changes ex vivo HIV infectivity in endocervical secretions. * *p* < 0.05. Values were bolded when the *p* value reached significance.

	Endocervical V2-V1		Endocervical V3-V1	
	Spearman r	*p* Value	Spearman r	*p* Value
**MCP-1**	0.1374	0.4771	0.1038	0.6064
**G-CSF**	0.1281	0.5160	−0.0939	0.6625
**TNF** α	0.0443	0.8163	0.0974	0.6358
**IL-6**	−0.2125	0.2597	0.1038	0.6064
**IL-1** β	−0.0328	0.8429	0.2666	0.1755
**IL-10**	−0.0995	0.6076	0.0147	0.9422
**GM-CSF**	0.0825	0.6646	0.2705	0.1724
**IL-2**	0.3222	0.0721	0.1441	0.4734
**MIP-1** α	−0.225	0.2236	0.0385	0.8489
**MIP-1** β	0.0407	0.8309	0.1392	0.4887
**IL-17**	**0.3833**	**0.0160 ***	−0.0562	0.7808
**IL-8**	−0.0928	0.6258	−0.1789	0.372
**IL-15**	0.1577	0.4052	−0.0702	0.7279
**IL-12p40**	0.3449	0.0620	−0.0147	0.9422
**VEGF**	**0.5164**	**0.0035 ***	0.2387	0.2304
**RANTES**	** − ** **0.6374**	**0.0003 ***	** − ** **0.4797**	**0.0131 ***
**IP-10**	0.1722	0.3545	0.1221	0.544
**IL-4**	**0.1278**	**0.0493 ***	0.1868	0.3508
**IL-7**	** − ** **0.3768**	**0.0439 ***	−0.0205	0.9193
**EOTAXIN**	0.2355	0.2188	−0.0263	0.8966
**IFN** γ	0.0742	0.7187	0.0006	0.9976
**TNF** β	0.0109	0.9544	−0.1093	0.5874
**IL-5**	−0.1111	0.545	−0.1273	0.5269
**IFN** α **2**	−0.0598	0.7534	−0.0965	0.6322
**IL-12p70**	0.119	0.532	−0.294	0.1538
**IL-13**	0.1862	0.3245	0.2885	0.1444
**IL-1** α	0.2176	0.1833	**0.3803**	**0.0500 ***
**EGF**	−0.0905	0.6342	−0.2239	0.2715

## Data Availability

Data are contained within the article.

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
