# Peer review of "Cervical Secretions from Women After Depot Medroxyprogesterone Acetate (Depo-Provera) Administration Promote HIV Infectivity Ex Vivo"

_viruses, 2025, doi:10.3390/v17091283_

Round 1
Reviewer 1 Report
Comments and Suggestions for Authors
The study of the connection between depot medroxyprogesterone acetate (DMPA) and HIV infectivity is of great importance from a broad clinical, immunological, and epidemiological perspective.
Considering that DMPA may have immunosuppressive effects by altering the activity of immune cells such as T cells, dendritic cells, and macrophages, which are key in the defense against HIV, and that it is one of the most widely used contraceptives in regions with high HIV prevalence, I believe that the article is highly relevant, methodologically well-conceived, and the statistical methods used are appropriate.
I believe it would be very important to expand and deepen the discussion sections of the results and conclusions, with greater emphasis on the role of each of the immune mediators in relation to HIV, mentioning the effect of DMPA on epithelial thinning, reduced production of protective cervical mucus, and alteration of the vaginal microbiome, and expanding the information regarding increased expression of CCR5 and CXCR4. It would also be important to infer about the variations found in immune mediators in v1 and v3.
DMPA is capable of modifying the concentration and function of IL-17 and VEGF, considered key immunological mediators in the interaction between the genital mucosa, local immunity, and HIV, which negatively affects the mucosa's ability to resist infections.
Short-term use could lead to a predominance of acute inflammation and the recruitment of target cells. In the long term, this could result in a persistent inflammatory environment, cumulative structural damage, and a weakening of the immune barrier.
Expanding the number of patients in future studies, taking ethnicity and some genetic factors into consideration would also be interesting, and, as mentioned, including patients with prolonged use of DMPA.
Author Response
Please see attached point-by-point file.

Reviewer 2 Report
Comments and Suggestions for Authors
The manuscript number viruses-3783543 entitled “Cervical secretions from women after Depot medroxyprogesterone acetate (Depo-Provera) administration promote HIV infectivity ex vivo” by Tasker et al. assesses the impact of Depo-provera on ex vivo HIV susceptibility in a cohort of women in USA between 18 and 35 years-old. This brief report is relevant considering the current discrepancies on the effects of Depo-provera use.
The authors should address the following points:
- The first sentence of the Introduction states that injectable Depo-provera “is associated with a higher risk of HIV acquisition”. However, multiple trials have shown inconsistent results. The authors should review this body of work and rephrase the introductory statement.
- Lines 43-54 and 56-57: This section describes published results and should be written in past tense, similarly to the sentence in lines 54-56.
- Was PSA measured to confirmed that participants did not have sexual intercourse for 3 days before enrollment visit?
- Levels of medroxyprogesterone acetate should have been measured, because some studies have shown a decrease of MPA concentration already at 2 months post-injection.
- Figure 1: the authors should discuss the significance observed in vaginal swabs between V3 and V2, and not when comparing V3 to V1. Is this significance linked to the data contraction at V2? Furthermore, the same data contraction is observed in data from endocervical swabs at V2.
Author Response
Please see the attached point-by-point file. Thanks.

Round 2
Reviewer 2 Report
Comments and Suggestions for Authors
The authors have addressed each comment appropriately.